# Elevated Cardiac Troponin I as a Mortality Predictor in Hospitalised COVID-19 Patients

**DOI:** 10.3390/medicina60060842

**Published:** 2024-05-21

**Authors:** Ieva Kubiliute, Jurgita Urboniene, Fausta Majauskaite, Edgar Bobkov, Linas Svetikas, Ligita Jancoriene

**Affiliations:** 1Clinic of Infectious Diseases and Dermatovenerology, Institute of Clinical Medicine, Faculty of Medicine, Vilnius University, 03101 Vilnius, Lithuania; fausta.majauskaite@santa.lt (F.M.); linas.svetikas@santa.lt (L.S.); ligita.jancoriene@santa.lt (L.J.); 2Centre of Infectious Diseases, Vilnius University Hospital Santaros Klinikos, 08661 Vilnius, Lithuania; jurgita.urboniene@santa.lt; 3Faculty of Medicine, Vilnius University, 03101 Vilnius, Lithuania; edgar.bobkov@santa.lt

**Keywords:** COVID-19, in-hospital mortality, cardiac troponin I, SARS-CoV-2

## Abstract

*Background and Objectives:* SARS-CoV-2 affects multiple organ systems, including the cardiovascular system, leading to immediate and long-term cardiovascular complications. Acute myocardial injury is one of the earliest and most common cardiac issues in the acute phase of COVID-19. This study aimed to evaluate the prognostic value of cardiac troponin I (cTnI) levels in predicting in-hospital mortality among hospitalised COVID-19 patients. *Materials and Methods:* A retrospective observational cohort study included 2019 adult patients hospitalised with a confirmed COVID-19 infection stratified by cTnI levels on admission into three groups: <19 ng/L (1416 patients), 19–100 ng/L (431 patients), and >100 ng/L (172 patients). Myocardial injury was defined as blood serum cTnI levels increased above the 99th percentile upper reference limit. Depersonalised datasets were extracted from digital health records. Statistical analysis included multivariable binary logistic and Cox proportional hazards regressions. *Results:* Overall, 29.87% of patients experienced acute myocardial injury, which development was associated with age, male sex, chronic heart failure, arterial hypertension, obesity, and chronic kidney disease. Among patients with cTnI levels of 19–100 ng/L, the odds ratio for requiring invasive mechanical ventilation was 3.18 (95% CI 2.11–4.79) and, for those with cTnI > 100 ng/L, 5.38 (95% CI 3.26–8.88). The hazard ratio for in-hospital mortality for patients with cTnI levels of 19–100 ng/L was 2.58 (95% CI 1.83–3.62) and, for those with cTnI > 100 ng/L, 2.97 (95% CI 2.01–4.39) compared to patients with normal cTnI levels. *Conclusions:* Increased cardiac troponin I, indicating myocardial injury, on admission is associated with a more adverse clinical disease course, including a higher likelihood of requiring invasive mechanical ventilation and increased risk of in-hospital mortality. This indicates cardiac troponin I to be a beneficial biomarker for clinicians trying to identify high-risk COVID-19 patients, choosing the optimal monitoring and treatment strategy for these patients.

## 1. Introduction

The COVID-19 pandemic, caused by severe acute respiratory syndrome coronavirus 2 (SARS-CoV-2), has emerged as a major global public health crisis. It appears that SARS-CoV-2 affects multiple organ systems, including the cardiovascular system, leading to both immediate and long-term cardiovascular complications. During the acute phase of infection, SARS-CoV-2 can cause direct cardiac injury by interacting with the angiotensin-converting enzyme 2 (ACE-2) receptor on various cardiac cells, such as cardiomyocytes, pericytes, macrophages, fibroblasts, and endothelial cells [1,2]. Autopsy studies of COVID-19 patients have confirmed the presence of the SARS-CoV-2 genome in cardiac tissues [3,4]. Furthermore, the immune response to SARS-CoV-2, resulting in a cytokine storm, may also cause direct cardiac damage. Indirect pathways of cardiac injury may involve myocardial ischemia, which could be precipitated by conditions like tachycardia, shock, severe respiratory distress (resembling type 2 myocardial infarction), or acute atherothrombotic incidents (resembling type 1 myocardial infarction). Clinical manifestations of cardiac injury include myocarditis, pericarditis, arrhythmia, myocardial infarction, heart failure, right ventricular dysfunction, and Takotsubo cardiomyopathy [5,6,7,8]. Acute myocardial injury, characterised by elevated serum cardiac troponin without the typical signs of acute ischemia on electrocardiograms or echocardiograms, is identified as one of the earliest and most common cardiac issues in the acute phase of COVID-19 [5].

This study aimed to evaluate the prognostic value of cardiac troponin I (cTnI) levels in predicting in-hospital mortality among hospitalised COVID-19 patients.

## 2. Materials and Methods

A retrospective observational cohort study was conducted at Vilnius University Hospital Santaros Klinikos in Lithuania from March 2020 to May 2021 [9].

The study included adult patients, 18 years and older, who were admitted to the hospital with confirmed COVID-19 infection and treated in various units, including standard care, high dependency, or intensive care units. The confirmation of the infection was based on a positive result on either a SARS-CoV-2 reverse transcriptase polymerase chain reaction or a rapid SARS-CoV-2 antigen test conducted on a nasopharyngeal sample. For symptomatic patients, the rapid antigen test was utilised within the first five days of the onset of COVID-19 symptoms. Initially, 2179 patients, who underwent cTnI testing upon admission, were considered for the study. Exclusion criteria included individuals with end-stage chronic kidney disease, defined by an estimated glomerular filtration rate (eGFR) less than 15 mL/min/1.73 m^2^, and those who developed myocardial infarction subsequent to admission. After applying these exclusions, the study population was finalised at 2019 patients. These individuals were stratified by cTnI levels on admission into three groups: 1416 patients exhibited cTnI levels below 19 ng/L, 431 patients had cTnI levels within the 19–100 ng/L range, and 172 patients presented with cTnI levels above 100 ng/L (Figure 1). Myocardial injury was defined as blood serum cTnI levels increased above the 99th percentile upper reference limit [10]. The concentration of cTnI was measured using the Snibe MAGLUMI 2000 Chemoluminescence Immunoassay System, Shenzhen New Industries Biomedical Engineering Co., Ltd., Shenzhen, China.

Depersonalised datasets were extracted from digital health records, facilitated by the Centre for Informatics and Development, in compliance with the procedures authorised by Vilnius University Hospital Santaros Klinikos. The demographic parameters encompassed sex and age. Comorbid conditions included arterial hypertension (AH), coronary artery disease (CAD), congestive heart failure (CHF), atrial fibrillation (AF), diabetes mellitus, obesity, chronic obstructive pulmonary disease (COPD), chronic kidney disease (CKD), and a history of cerebrovascular accidents. These conditions were identified within the electronic medical records using the corresponding International Classification of Diseases, Tenth Revision (ICD-10) codes. If a medical condition was not documented in the electronic medical records related to the hospitalisation due to COVID-19, the individual was classified as not having that specific comorbidity.

Initial laboratory test results, including a complete blood count, creatinine, urea, eGFR, sodium, potassium, alanine aminotransferase (ALT), aspartate aminotransferase (AST), lactate dehydrogenase (LDH), C-reactive protein (CRP), ferritin, interleukin 6 (IL-6), D-dimer, and cTnI levels, were extracted from electronic medical records and analysed. Additionally, information regarding the duration of hospitalisation and patient outcomes was gathered from depersonalised medical records.

Statistical analysis was performed using IBM SPSS version 20.0. Continuous and categorical variables were presented as the median (interquartile range (IQR)) and numbers (percentages), respectively. Mann–Whitney *U* test was used to compare continuous variables, and χ^2^ test or Fisher’s exact test was used to compare categorical variables. Multivariable binary logistic regression was used to determine the association of myocardial injury development with age, sex, and comorbid conditions. Another multivariable binary logistic regression model was created to determine the association of cTnI levels with the need for invasive mechanical ventilation (IMV). The model included the need for IMV as the dependent variable and age; sex; and comorbid conditions (AH, CAD, CHF, AF, diabetes, obesity, COPD, CKD, and previous stroke) as predictors. We conducted a survival analysis to assess the effects of cTnI levels on outcomes within 30 days of hospitalisation. Patients were considered to be right-censored if they were discharged from the hospital alive or remained in the hospital. Cox proportional hazards regression models were created with in-hospital mortality as the dependent variable and age; comorbid conditions (AH, CAD, CHF, AF, diabetes, obesity, COPD, CKD, and previous stroke); treatment with remdesivir; systemic steroids; and antibiotics as predictors. Survival curves were plotted to illustrate the survival rates stratified by the cTnI groups. A two-sided *p*-value <0.05 was considered significant.

## 3. Results

Among 2019 adults, 56% were men. The median age was 60 years (IQR 50–69). Table 1 presents the demographic details, clinical features, and initial laboratory findings of the patients included in the analysis.

Out of 2019 patients, 603 (29.87%) experienced acute myocardial injury, characterised by elevated levels of cTnI. Multivariable regression analysis revealed that age, male sex, the presence of CHF, AF, obesity, and CKD were associated with the development of acute myocardial injury (Figure 2).

Out of the 603 patients with elevated cTnI levels, 431 (71.48%) had cTnI levels between 19 and 100 ng/L, and 172 (28.52%) had cTnI levels above 100 ng/L. Patients with elevated cTnI levels were, on average, older than those presenting with normal cTnI levels upon admission. The prevalence of any comorbid condition, including AH, CAD, CHF, AF, diabetes, COPD, CKD, and previous stroke, was higher among patients with cTnI levels of 19–100 ng/L and in those with cTnI levels above 100 ng/L compared to patients with normal cTnI levels. The prevalence of CAD and AF was greater in patients with cTnI levels of 19–100 ng/L than in those with cTnI levels above 100 ng/L (Table 2).

Patients with cTnI levels of 19–100 ng/L exhibited statistically significantly higher levels of WBC, neutrophil count, glucose, creatinine, urea, AST, LDH, CRP, ferritin, IL-6, and D-dimer compared to patients with normal cTnI levels. Conversely, their lymphocyte count, eGFR, and sodium concentration were significantly lower (Table 3).

Patients with cTnI levels above 100 ng/L showed higher levels of WBC, neutrophil count, glucose, creatinine, urea, AST, LDH, and D-dimer compared to both patients with normal cTnI levels and those with cTnI levels of 19–100 ng/L. CRP, ferritin, and IL-6 concentrations were higher in patients with cTnI levels above 100 ng/L than in those with normal cTnI levels; however, there was no statistically significant difference in CRP, ferritin, and IL-6 concentrations between patients with cTnI levels above 100 ng/L and those with levels of 19–100 ng/L. Lymphocyte count, eGFR, and sodium concentration were significantly lower in patients with cTnI levels above 100 ng/L compared to those with normal cTnI levels (Table 3).

Remdesivir was administered to 711 (35.22%) patients upon admission or the following day for a standard 5-day course (IQR 5 days). Patients with normal cTnI levels and those with cTnI levels of 19–100 ng/L were more likely to receive remdesivir than those with cTnI levels above 100 ng/L at 36.23% and 37.59%, respectively, compared to 20.93% (*p* < 0.001 for both comparisons).

Systemic steroids were administered to 1371 (67.90%) patients, with the majority (97.67%) being treated with dexamethasone for a median duration of 9 days (IQR 6–10 days). Fewer patients with cTnI levels greater than 100 ng/L received steroids compared to those with cTnI levels of 19–100 ng/L (55.23% vs. 73.09%, *p* < 0.001) and those with normal cTnI levels (55.23% vs. 67.87%, *p* = 0.001).

A total of 1514 (74.99%) patients received antibiotics. The most commonly administered antibiotics were amoxicillin and clavulanic acid, which were used in 1269 of the 1514 cases (83.82%). Other antibiotics were prescribed less frequently—piperacillin and tazobactam—in 454 cases (29.99%) and azithromycin in 120 cases (7.93%). Antibiotic use was less frequent in patients with cTnI levels greater than 100 ng/L compared to those with cTnI levels of 19–100 ng/L (65.12% vs. 77.96%, *p* = 0.001) and also compared to patients with normal cTnI levels (65.12% vs. 77.96%, *p* = 0.004).

The need for IMV was highest in patients with cTnI levels above 100 ng/L at 25% compared to 15.31% in those with cTnI levels between 19 and 100 ng/L, and 4.31% in patients with normal cTnI levels (Table 2). Among patients with cTnI 19–100 ng/L, the odds ratio (OR) for IMV was 3.18 (95% CI 2.11–4.79, *p* < 0.001); in those with cTnI levels above 100 ng/L, the OR was 5.38 (95% CI 3.26–8.88, *p* < 0.001) compared to patients presenting normal cTnI levels on admission. Other risk factors for IMV were male sex (OR 1.49, 95% CI 1.04–2.12, *p* = 0.029) and CHF (OR 2.85, 95% CI 1.82–4.45, *p* < 0.001).

The in-hospital mortality rate was highest among patients with cTnI levels above 100 ng/L at 37.79% compared to 24.59% in those with cTnI levels of 19–100 ng/L and 5.37% in patients with normal cTnI levels.

Cox regression analysis revealed that the hazard ratio (HR) for 30-day in-hospital mortality for patients with cTnI levels of 19–100 ng/L upon admission was 2.58 (95% CI 1.83–3.62, *p* < 0.001), and for those with cTnI levels above 100 ng/L, the HR was 2.97 (95% CI 2.01–4.39, *p* < 0.001) compared to those with normal cTnI levels (Figure 3).

Other risk factors associated with an increased HR for 30-day in-hospital mortality included age, CHF, obesity, and previous stroke. Treatment with remdesivir was associated with a reduced HR for 30-day in-hospital mortality: HR 0.61 (95% CI 0.43–0.86, *p* = 0.005) (Table 4).

## 4. Discussion

Several viral infections are identified as potential causes of heart damage. These include enteroviruses (Coxsackie B viruses, echoviruses, and poliovirus); herpesviruses (human herpes virus 6, Epstein–Barr virus, cytomegalovirus, and varicella zoster virus); parvoviruses (such as parvovirus B19); adenoviruses; influenza viruses; retroviruses (human immunodeficiency virus (HIV)); and coronaviruses [11,12]. In addition, accumulating data suggest that hepatitis C virus may also contribute to heart diseases [13]. Viruses can damage the heart through various mechanisms, including direct virus-mediated damage to cardiomyocytes (Coxsackievirus B, Epstein–Barr virus, cytomegalovirus, and HIV); endothelial cells (human herpes virus 6, parvovirus B19, and cytomegalovirus); and fibroblasts (cytomegalovirus). During the acute phase of infection, SARS-CoV-2 can cause direct cardiac injury by interacting with the ACE-2 receptor on cardiomyocytes, pericytes, macrophages, fibroblasts, and endothelial cells [1,2]. Moreover, viruses can cause heart damage through an immune response, which may result from the development of autoimmune myocarditis or excessive cytokine production [11]. In our research, we focused on myocardial injury caused by SARS-CoV-2 and analysed the data of one of the largest hospitals in Lithuania from March 2020, when the first COVID-19 patient was admitted to the hospital, until May 2021. In the initial stages of the pandemic, until the autumn of 2020, no single SARS-CoV-2 lineage clearly dominated in Lithuania. By the end of 2020, the B.1.177.60 lineage began to spread. Subsequently, this lineage was overtaken by the B.1.1.7 (alpha variant) and its localised version Q.1., which dominated until June 2021 [14].

In our single-centre retrospective cohort study of 2019 hospitalised COVID-19 patients, several key observations were made: (1) myocardial injury was quite common among patients hospitalised with COVID-19, affecting nearly 30% of patients, but in the majority of cases, it was associated with low-level elevations in cTnI concentrations; (2) age, male sex, the presence of CHF, AF, obesity, and CKD were associated with the development of acute myocardial injury; (3) patients with acute myocardial injury exhibited higher levels of inflammatory biomarkers; and (4) more significant myocardial injury was associated with a five-fold higher risk of requiring invasive mechanical ventilation and nearly a three-fold higher risk of in-hospital mortality. 

The detection of elevated myocardial injury markers, particularly cTnI, has garnered considerable scientific interest due to its widespread measurement in various healthcare settings and its clinical significance [15,16,17,18]. Numerous studies have reported myocardial injury, defining it as troponin elevation above the 99th percentile of the upper reference limit, with varying prevalence rates based on the population studied. For instance, Lala et al., in a study involving 2736 hospitalised COVID-19 patients, found a 36% prevalence of myocardial injury [16]. De Marzo et al. reported a myocardial injury prevalence of 25.7% among individuals exceeding 60 years of age [17], while Shi et al. documented myocardial injury in 19.7% of their study cohort [18]. In alignment with these findings, our research demonstrated that 29.87% of patients admitted to hospital for COVID-19 infection presented with increased cTnI levels. Despite varying prevalence rates of myocardial injury reported across studies, there is a consensus that myocardial injury occurs more frequently in severe or critically ill COVID-19 patients and non-survivors [19,20,21].

Furthermore, myocardial injury has been found to occur more frequently among older patients with a greater number of comorbidities, particularly chronic cardiovascular conditions [16,18,22,23]. Lala et al. highlighted that patients with more severe myocardial injury had a higher prevalence of cardiovascular diseases, such as coronary artery disease, atrial fibrillation, and heart failure, compared to those with mildly elevated or normal troponin levels [16]. Our research further quantified this risk, showing that age increased the odds ratio for myocardial injury by 1.06 times, male sex by 1.58 times, the presence of CHF by 2.79 times, AF by 1.72 times, obesity by 1.84 times, and CKD by 2.19 times. Similarly, Papageorgiou et al. identified that age (OR 1.68), hypertension (OR 1.81), and moderate CKD (OR 9.12) were independent predictors of myocardial injury [23]. Additionally, Lombardi and colleagues found significant associations between myocardial injury with heart failure (OR 2.01) and a history of coronary artery disease (OR 2.04) [22]. The existence of numerous comorbidities can exacerbate the severity of COVID-19, resulting in more severe myocardial injury.

Our study revealed that patients with myocardial injury presented with elevated levels of WBC, neutrophil count, glucose, creatinine, urea, AST, LDH, CRP, ferritin, IL-6, and D-dimer in comparison to those with normal cTnI levels. This correlation between myocardial injury and an increase in inflammatory and prothrombotic markers is consistent with findings from other researchers. Specifically, myocardial injury patients demonstrated significant rises in inflammatory biomarkers such as CRP [18,22,24] and thrombotic indicators like D-dimer [22,23,24]. Moreover, Li et al. identified correlations between high sensitivity cTnI levels and IL-6 (r = 0.59 to 0.66, *p* < 0.001), CRP (r = 0.54 to 0.62, *p* < 0.001), and D-dimer (r = 0.54 to 0.65, *p* < 0.001) at various stages: upon admission, within the first week, and after the first week of hospitalisation, further underscoring the link between cardiac injury and systemic inflammation and thrombosis in this patient population [20].

In our study, 35% of patients received treatment with remdesivir, and this treatment was associated with a 1.64-fold reduction in HR for 30-day in-hospital mortality. These findings are consistent with prior research, indicating that remdesivir administration is linked to improved time to recovery and potentially reduced mortality rates [25]. However, this finding should be approached with caution, as remdesivir was more frequently administered to patients with less severe COVID-19 and less severe acute myocardial injury, taking into consideration the reported potential cardiac effects associated with its use, such as hypotension, bradycardia, and prolonged QTc interval, that are more frequently observed in patients with COVID-19 and concomitant cardiovascular diseases [7,26,27]. Randomised controlled trials have provided strong evidence supporting the beneficial therapeutic effect of corticosteroids in COVID-19 patients requiring oxygen therapy, with no impact on cardiovascular events [28,29]. Our study demonstrated that patients with more severe acute myocardial injury were less frequently administered systemic steroids and that treatment with systemic steroids had no effect on the HR for 30-day mortality.

Our study revealed a 3.18-fold increase in the odds ratio for IMV and a 2.58-fold increase in the hazard ratio for 30-day in-hospital mortality among patients with cTnI levels between 19 and 100 ng/L. Higher cTnI levels correlated with even greater increases, up to 5.38 times for the OR for IMV and up to 2.97 times for the HR for in-hospital mortality. These findings are in close agreement with those of Lala et al., who observed a 3.03-fold increase in the HR for mortality (95% CI 2.42–3.80) in patients with troponin levels more than three times the upper reference limit, although in an older study population with higher incidences of coronary artery disease (CAD) and heart failure at 16.6% and 10.1%, respectively [16]. Similarly, Shi et al. noted a higher mortality risk in patients with myocardial injury, with a HR of 3.41 [18]. Papageorgiou et al. identified that patients with elevated troponin levels were significantly more likely to require non-invasive and IMV (with ORs of 2.40 and 6.81, respectively) to develop acute kidney injury (with 6.76-times higher odds), to necessitate urgent renal replacement therapy (with 4.14-times higher odds), to experience thromboembolic events (with 11.99-times higher odds), and to die (with 2.40-times higher odds) compared to patients with normal troponin levels [23].

The role of elevated cTnI levels as a critical prognostic marker for severe outcomes in COVID-19, including the necessity for IMV and higher mortality rates, has sparked discussions on its potential integration into routine diagnostics and its implications for tailoring treatment approaches for patients with myocardial injury [30,31]. Sandoval et al. proposed a structured method for leveraging cTnI in risk stratification. An initial cTnI measurement could be instrumental in the early classification of COVID-19 patients, aiding in their preliminary assessment. Patients deemed clinically stable or marginally stable yet presenting with significant laboratory anomalies such as elevated cTnI levels are categorised at an intermediate risk level. Conversely, patients who are clinically unstable with notable laboratory abnormalities fall into the high-risk category and necessitate hospital admission. For hospitalised patients, repeated cTnI monitoring is recommended. Individuals whose cTnI levels remain stable and do not exhibit significant increases might not require further interventions. On the other hand, patients exhibiting escalating cTnI levels should be subjected to more in-depth evaluations and potentially more intensive therapeutic interventions [19].

We acknowledge that our study has several limitations. Firstly, the identification of comorbidities relied on ICD-10-AM codes encoded in medical records, which might not have captured all relevant conditions. Another issue was the absence of data regarding the duration of COVID-19 symptoms prior to hospitalisation, which prevented the specification of the time of myocardial injury and the more precise evaluation of the potential mechanism of the myocardial injury in regard to its relation to virus-mediated or immune-mediated damage and evolution over time. The dataset did not include the dates of hospitalisation, which did not allow for the analysis of acute myocardial damage across various COVID-19 pandemic waves. Conducted in the largest hospital of the capital of Lithuania, the single-centre nature of this study may lead to selection bias and limit the applicability of the results to wider populations. The specific healthcare practices, patient demographics, and treatment approaches of this hospital may not accurately reflect those of the broader population. Furthermore, our retrospective study’s design did not allow for the assessment of the long-term effects of acute myocardial injury in COVID-19 patients, as it focused on short-term outcomes during the hospital stay and not addressing the potential prolonged impact of acute myocardial injury beyond the acute phase of the infection.

## 5. Conclusions

Cardiac troponin I serves as an effective biomarker for risk stratification in COVID-19 patients upon admission. Increased levels of cardiac troponin I, indicating myocardial injury, on admission are associated with a more adverse clinical disease course, including a higher likelihood of the need for invasive mechanical ventilation and increased risk of in-hospital mortality. This indicates cardiac troponin I to be a beneficial biomarker for clinicians trying to identify high-risk COVID-19 patients, choosing the optimal monitoring and treatment strategy for these patients.

## Figures and Tables

**Figure 1 medicina-60-00842-f001:**
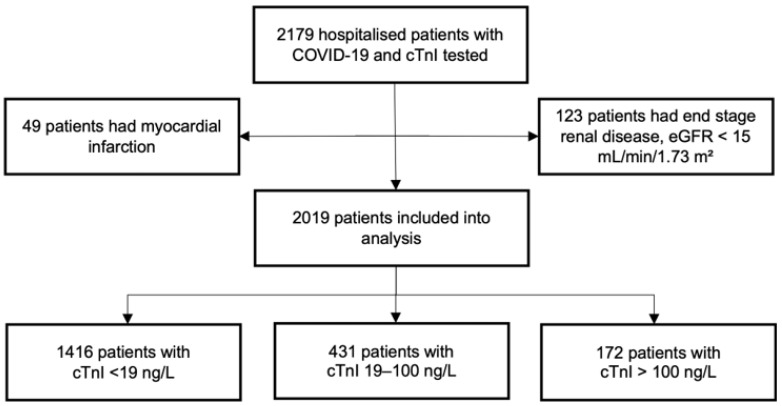
Distribution of patients hospitalised with COVID-19 infection.

**Figure 2 medicina-60-00842-f002:**
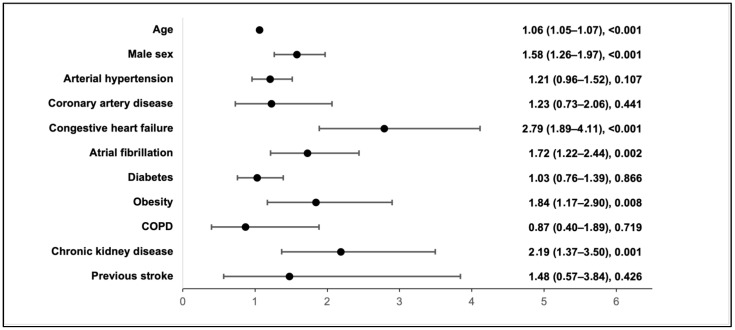
Risk factors and odds ratio (OR) associated with acute myocardial injury in COVID-19 patients.

**Figure 3 medicina-60-00842-f003:**
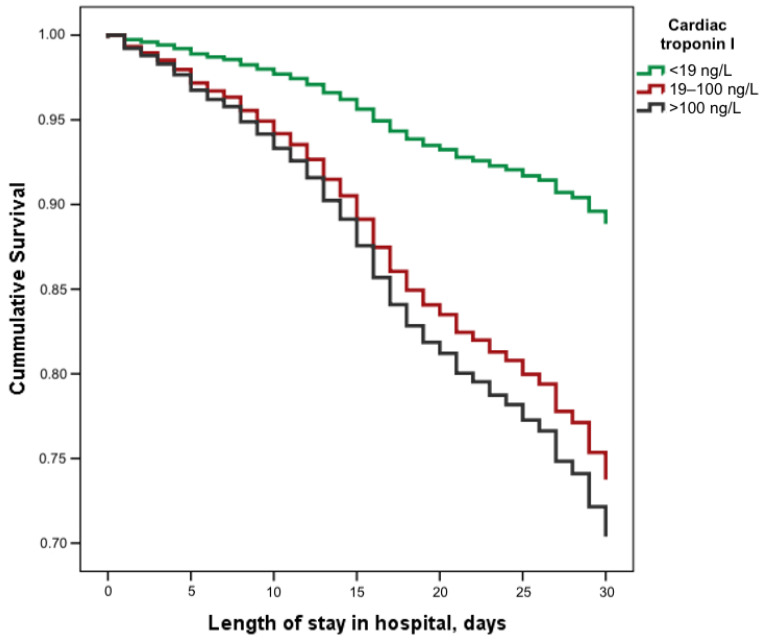
Survival of hospitalised COVID-19 patients stratified by cardiac troponin I concentration.

**Table 1 medicina-60-00842-t001:** Demographic, clinical, and initial laboratory characteristics of hospitalised COVID-19 patients.

Demographic and Clinical Characteristic (N = 2019)	N (%)	Laboratory Characteristics	N	Median (IQR)
Age, years, median (IQR)	60 (50–69)	Haemoglobin, g/L	2019	140.0 (127.0–150.0)
Male	1130 (55.97)	WBC, ×10^9^/L	2019	6.34 (4.76–8.61)
Female	889 (44.03)	Neutrophils, ×10^9^/L	2019	4.66 (3.28–6.69)
Any comorbid condition	1000 (49.53)	Lymphocytes, ×10^9^/L	2019	1.0 (0.71–1.40)
Arterial hypertension	802 (39.72)	Platelets, ×10^9^/L	2019	195.0 (153.0–253.0)
Coronary artery disease	79 (3.91)	cTnI	2019	9.70 (5.0–23.0)
Congestive heart failure	160 (7.92)	Glucose, mmol/L	1984	6.22 (5.54–7.30)
Atrial fibrillation	203 (10.05)	Creatinine, µmol/L	2019	81.0 (67.0–100.0)
Diabetes mellitus	277 (13.72)	eGFR, mL/min/1.73 m^2^	2019	84.0 (64.0–96.0)
Obesity	106 (5.25)	Urea, mmol/L	1883	5.50 (4.07–7.91)
COPD	34 (1.68)	Sodium, mmol/L	2019	140.0 (137.0–143.0)
Chronic kidney disease	109 (5.40)	Potassium, mmol/L	2019	4.20 (3.87–4.50)
Previous stroke	23 (1.14)	ALT, U/L	1993	33.0 (21.0–53.0)
Invasive mechanical ventilation	170 (8.42)	AST, U/L	1986	37.0 (26.98–57.0)
Treatment with antibiotics	1514 (74.99)	LDH, U/L	1910	308.0 (243.0–410.0)
Treatment with antivirals (Remdesivir)	711 (35.22)	CRP, mg/L	2018	63.25 (25.45–125.53)
Treatment with systemic steroids	1371 (67.90)	Ferritin, µg/L	1939	471.0 (235.00–1 002.0)
In-hospital mortality	247 (12.23)	IL-6, ng/L	1892	30.40 (15.01–57.30)
Length of hospital stay, days, median (IQR)	11 (8–16)	D-dimer, µg/L	1972	490.0 (295.0–910.0)

ALT: alanine aminotransferase; AST: aspartate aminotransferase; COPD: chronic obstructive pulmonary disease; CRP: C-reactive protein; cTnI: cardiac troponin I; eGFR: estimated glomerular filtration rate; IL-6: interleukin 6; IQR: interquartile range; LDH: lactate dehydrogenase; N: number; WBC: white blood cell count. Reference values: haemoglobin: 128–160 g/L (for males), 117–145 g/L (for females); WBC: 4.0–9.8 ×10^9^/L; neutrophils: 1.5–6.0 ×10^9^/L; lymphocytes: 1.0–4.0 ×10^9^/L; NLR: 1–2; platelets: 140–450 ×10^9^/L; glucose: 4.2–6.1 mmol/L; creatinine: 64–104 µmol/L (for males), 49–90 µmol/L (for females); urea: 2.5–7.5 mmol/L; sodium: 134–145 mmol/L; potassium: 3.8–5.3 mmol/L; ALT: ≤40 U/L; AST: ≤40 U/L; LDH: 125–243 U/L; CRP: <5 mg/L; ferritin: 25–350 µg/L (for men), 13–232 µg/L (for women); IL-6: 0–7 ng/L; D-dimer: <250 µg/L.

**Table 2 medicina-60-00842-t002:** Baseline characteristics of hospitalised patients stratified by cardiac troponin I concentration.

Characteristic	cTnI < 19 ng/L (N = 1416), *n* (%)	19 ≤ cTnI ≤ 100 ng/L (N = 431), *n* (%)	cTnI > 100 ng/L (N = 172), *n* (%)	*p*-Value ^1^	*p*-Value ^2^	*p*-Value ^3^
Age in years, median (IQR)	57 (47–65)	67 (58–77)	74 (62–81)	<0.001	<0.001	<0.001
Male	771 (54.45)	253 (58.70)	106 (61.63)	0.120	0.074	0.508
Any underlying condition	584 (41.24)	287 (66.59)	129 (75.00)	<0.001	<0.001	0.044
Hypertension	481 (33.97)	220 (51.04)	101 (58.72)	<0.001	<0.001	0.088
Coronary artery disease	33 (2.33)	27 (6.26)	19 (11.05)	<0.001	<0.001	0.046
Congestive heart failure	50 (3.53)	72 (16.71)	38 (22.09)	<0.001	<0.001	0.122
Atrial fibrillation	74 (5.23)	81 (18.79)	48 (27.91)	<0.001	<0.001	0.014
Diabetes	160 (11.30)	82 (19.03)	35 (20.35)	<0.001	0.001	0.711
Obesity	63 (4.45)	29 (6.73)	14 (8.14)	0.057	0.033	0.543
COPD	17 (1.20)	11 (2.55)	6 (3.49)	0.044	0.031	0.587
Chronic kidney disease	37 (2.61)	46 (10.67)	26 (15.12)	<0.001	<0.001	0.129
Previous stroke	7 (0.49)	10 (2.32)	6 (3.49)	0.002	0.001	0.410
Complications						
Pulmonary embolism	20 (1.41)	17 (3.94)	16 (9.30)	0.001	<0.001	0.009
Stroke	12 (0.85)	15 (3.48)	3 (1.74)	<0.001	0.216	0.258
Remdesivir	513 (36.23)	162 (37.59)	36 (20.93)	0.608	<0.001	<0.001
Systemic steroids	961 (67.87)	315 (73.09)	95 (55.23)	0.040	0.001	<0.001
Antibiotics use	1066 (75.28)	336 (77.96)	112 (65.12)	0.255	0.004	0.001
Invasive ventilation	61 (4.31)	66 (15.31)	43 (25.00)	<0.001	<0.001	0.005
Length of hospital stay, days	11 (7–15)	12 (8–19)	13 (8–21)	<0.001	0.001	0.579
In-hospital mortality	76 (5.37)	106 (24.59)	65 (37.79)	<0.001	<0.001	0.001

(1) *p*-value comparing patients with cTnI < 19 ng/L vs. patients with 19 ≤ cTnI ≤ 100 ng/L. (2) *p*-value comparing patients with cTnI < 19 ng/L vs. patients with cTnI > 100 ng/L. (3) *p*-value comparing patients with 19 ≤ cTnI ≤ 100 ng/L vs. patients with cTnI > 100 ng/L. cTnI: cardiac troponin I; IQR: interquartile range; N or n: number.

**Table 3 medicina-60-00842-t003:** Laboratory tests of hospitalised COVID-19 patients stratified by cardiac troponin I concentration.

Variable	cTnI < 19 ng/L (N = 1416), Median (IQR)	19 ≤ cTnI ≤ 100 ng/L (N = 431), Median (IQR)	cTnI > 100 ng/L (N = 172), Median (IQR)	*p*-Value ^1^	*p*-Value ^2^	*p*-Value ^3^
Haemoglobin, g/L	142.00 (130.00–151.00)	135.00 (121.00–147.00)	133.00 (116.00–145.75)	<0.001	<0.001	0.164
WBC, ×10^9^/L	6.15 (4.65–8.06)	6.72 (5.02–9.41)	8.83 (6.08–12.38)	<0.001	<0.001	<0.001
Neutrophils, ×10^9^/L	4.42 (3.14–6.15)	4.95 (3.50–7.53)	7.10 (4.38–10.23)	<0.001	<0.001	<0.001
Lymphocytes, ×10^9^/L	1.04 (0.77–1.40)	0.90 (0.60–1.39)	0.80 (0.53–1.30)	<0.001	<0.001	0.071
Platelets, ×10^9^/L	195.00 (154.00–252.00)	192.00 (151.00–245.00)	207.00 (150.25–261.25)	0.202	0.495	0.195
Glucose, mmol/L	6.07 (5.46–6.92)	6.67 (5.82–8.23)	7.64 (6.02–9.76)	<0.001	<0.001	0.003
Creatinine, µmol/L	78.00 (65.00–91.00)	92.00 (72.00–125.00)	103.50 (77.00–148.75)	<0.001	<0.001	0.018
eGFR, mL/min/1.73 m^2^	88.00 (73.08–98.30)	68.40 (43.60–89.00)	57.20 (34.00–82.00)	<0.001	<0.001	0.002
Urea, mmol/L	4.91 (3.74–6.48)	7.37 (5.24–11.14)	10.17 (6.68–16.25)	<0.001	<0.001	<0.001
Sodium, mmol/L	140.75 (137.88–143.00)	140.00 (136.10–143.00)	139.00 (135.00–142.00)	0.001	0.002	0.386
Potassium, mmol/:	4.15 (3.90–4.50)	4.20 (3.80–4.60)	4.20 (3.85–4.60)	0.468	0.075	0.314
ALT, U/L	33.00 (21.00–53.00)	31.00 (19.35–50.00)	32.50 (19.00–59.75)	0.066	0.676	0.552
AST, U/L	35.00 (25.72–53.00)	40.00 (29.08–61.07)	50.00 (31.00–78.75)	<0.001	<0.001	0.004
LDH, U/L	295.00 (236.00–388.00)	335.00 (264.00–487.00)	385.50 (280.75–564.75)	<0.001	<0.001	0.031
CRP, mg/L	59.00 (24.43–108.90)	85.60 (32.60–157.60)	92.08 (19.88–146.30)	<0.001	0.003	0.486
Ferritin, µg/L	441.00 (225.50–952.55)	543.00 (252.50–1115.50)	543.29 (239.70–1286.73)	0.011	0.046	0.739
IL-6, ng/L	27.70 (13.90–51.05)	42.50 (19.85–71.05)	34.15 (16.85–87.73)	<0.001	0.001	0.739
D-dimer, µg/L	430.00 (270.00–720.00)	677.50 (410.00–1330.00)	1 255.00 (587.50–2360.00)	<0.001	<0.001	<0.001

ALT: alanine aminotransferase; AST: aspartate aminotransferase; CRP: C-reactive protein; cTnI: cardiac troponin I; eGFR: estimated glomerular filtration rate; IL-6: interleukin 6; IQR: interquartile range; LDH: lactate dehydrogenase; N: number; WBC: white blood cell count. (1) *p*-value comparing cTnI < 19 ng/L vs. 19 ≤ cTnI ≤ 100 ng/L. (2) *p*-value comparing cTnI < 19 ng/L vs. cTnI > 100 ng/L. (3) *p*-value comparing 19 ≤ cTnI ≤ 100 ng/L vs. cTnI > 100 ng/L. Reference values: haemoglobin: 128–160 g/L (for males), 117–145 g/L (for females); WBC: 4.0–9.8 ×10^9^/L; neutrophils: 1.5–6.0 ×10^9^/L; lymphocytes: 1.0–4.0 ×10^9^/L; platelets: 140–450 ×10^9^/L; glucose: 4.2–6.1 mmol/L; creatinine: 64–104 µmol/L (for males), 49–90 µmol/L (for females); eGFR: >90 mL/min/1.73 m^2^; urea: 2.5–7.5 mmol/L; sodium: 134–145 mmol/L; potassium: 3.8–5.3 mmol/L; ALT: ≤40 U/L; AST: ≤40 U/L; LDH: 125–243 U/L; CRP: <5 mg/L; ferritin: 25–350 µg/L (for men), 13–232 µg/L (for women); IL-6: 0–7 ng/L; D-dimer: <250 µg/L.

**Table 4 medicina-60-00842-t004:** Hazards ratio for 30 days in-hospital mortality in hospitalised COVID-19 patients.

Characteristic	In-Hospital Mortality	
	HR (95% CI)	*p*-Value
TnI < 19 ng/L	Reference	
19 ≤ TnI ≤ 100 ng/L	2.58 (1.83–3.62)	<0.001
TnI > 100 ng/L	2.97 (2.01–4.39)	<0.001
Age in years	1.05 (1.03–1.06)	<0.001
Arterial hypertension	0.71 (0.53–0.95)	0.019
Coronary artery disease	1.03 (0.63–1.67)	0.906
Congestive heart failure	1.81 (1.33–2.46)	<0.001
Atrial fibrillation	1.45 (1.06–1.98)	0.021
Diabetes	0.92 (0.66–1.29)	0.636
Obesity	2.40 (1.50–3.84)	<0.001
COPD	1.75 (0.96–3.18)	0.068
Chronic kidney disease	0.70 (0.46–1.05)	0.083
Previous stroke	1.67 (0.88–3.16)	0.114
Treatment with remdesivir	0.61 (0.43–0.86)	0.005
Treatment with systemic steroids	1.40 (1.00–1.96)	0.051
Antibiotics	0.85 (0.59–1.24)	0.413

COPD: chronic obstructive pulmonary disease; cTnI: cardiac troponin I; HR: hazard ratio.

## Data Availability

The data presented in this study are available on request from the corresponding author.

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
