# Peer review of "Elevated Cardiac Troponin I as a Mortality Predictor in Hospitalised COVID-19 Patients"

_medicina, 2024, doi:10.3390/medicina60060842_

Round 1

Reviewer 1 Report

Comments and Suggestions for Authors

This is a nice written paper and the results presented by the investigators are in line with what we know so far. Nevertheless, the originality of the subject is lacking.

We already know that myocardial injury is associated with a worse prognosis during COVID-19 pneumonia. A more interesting subject would have been to show if the viral strain plays an important role in the myocardial injury severity and prevalence in COVID-19 patiens.

Positive points: large cohort of patients

Negative points: lack of originality, lack of information regarding the treatment used during COVID-19 and the impact on patients' prognosis. The authors investigated various COVID-19 waves. It would be interesting if there is a variation in time based on the COVID-19 wave and the prevalence and severity of myocardial injury. Moreover, the vaccination status evolved during the COVID-19 waves . This is an important point tacking into consideration the wordlwide debate on COVID-19 vaccination side effects and its beneficial effects.What about the cardiac function of those patients with myocardial injury?

 Second point: discussion paragraph - too short. More uptodated references shout be used. Is there a difference between myocardial injury in COVID-19 versus other viral diseases? Is there a difference in terms of vaccination and COVID-19 prognosis and outcomes. What about 2024 data on COVID-19? What about long covid?

There is no mention about future perspectives, research..

In conclusion, this is a nice study, coming to reinforce the data we have so far on myocardial injury in COVID-19. The results are in line with what we know. They used a large cohort of patients to prove the importance of cardiac troponin as a prognosis biomarker. It will not change current clinical practice. 

Reviewer 2 Report

Comments and Suggestions for Authors

I appritiate the opportunity to read and review this article. The major strength of this study is affort to predict mortality at the very begining of the disease. The major limitations are the lack of clinical data and their comparisons to each other and with age, sex, obesity, and other well-known risk factors (as a confounding bias).

1. "These individuals were stratified by cTnI levels on admission into three 71 groups: 1,416 patients exhibited cTnI levels below 19 ng/L, 431 patients had cTnI within 72 the 19–100 ng/L range, and 172 patients presented with cTnI levels above 100 ng/L"  - Could autors describe how many days were left after the symptoms onset?

The higher level of troponin could be at more sick patients.

2. Clinical data are not enough. Were there any arthery disease (e.g. stenosis of lower limb arteries)? it could be another cause of elevated troponin level

3. What was the treatment of suspected myocardial injury? 
